



# Understanding representations of uncertainty, an eye-tracking study part I: The effect of anchoring

Kelsey J. Mulder[1,2], Louis Williams[3], Matthew Lickiss[4], Alison Black[4], Andrew Charlton-Perez[1], Rachel McCloy[3], and Eugene McSorley[3]

[1]Department of Meteorology, University of Reading
[2](Current Affiliation) Liberty Specialty Markets, 20 Fenchurch Street, London EC3M 3AW, UK
[3]Department of Psychology, University of Reading
[4]Department of Typography and Graphic Communication, University of Reading

**Correspondence:** Kelsey J. Mulder (kelsey.mulder@libertyglobalgroup.com)

**Abstract.** Geoscience communicators must think carefully about how uncertainty is represented and how users may interpret these representations. Doing so will help communicate risk more effectively, which can elicit appropriate responses. Recently, communication of uncertainty has come to the forefront over the course of the COVID-19 pandemic, but the lessons learned from communication during the pandemic can be adopted across geosciences as well. To test interpretations of environmental
forecasts with uncertainty, a decision task survey was administered to 65 participants who saw different hypothetical forecast representations common to presentations of environmental data and forecasts: deterministic, spaghetti plot with and without a median line, fan plot with and without a median line, and box plot with and without a median line. While participants completed the survey, their eye movements were monitored with eye-tracking software. Participants' eye movements were anchored to the median line, not focusing on possible extreme values to the same extent as when no median line was present. Additionally,
participants largely correctly interpreted extreme values from the spaghetti and fan plots, but misinterpreted extreme values from the box plot, perhaps because participants spent little time fixating on the key. These results suggest that anchoring lines, such as median lines, should only be used where users should be guided to particular values and where extreme values are not as important in data interpretation. Additionally, fan or spaghetti plots should be considered instead of box plots to reduce misinterpretation of extreme values. Further study on the role of expertise and the change in eye movements across the graph
area and key is explored in more detail in the companion paper to this study.

## 1  Introduction

Over the course of the COVID-19 pandemic, the last-mile of the forecasting process, the need to effectively and clearly communicate forecasts and their inherent uncertainty has been brought into sharp focus. Real-world decisions by members of the public with no specialist training and that have major public health, social and economic impacts depend on this last-mile.
Without careful design of how forecasts are communicated, especially with respect to uncertainty, runs the risk of misinterpretation and more importantly poorly informed decision making that exacerbates the impact of the forecast hazard on those affected. In the environmental science, there are numerous examples of the need to communicate uncertain forecasts to the





public. These include forecasts of short-term hazards like landfalling hurricanes and property flooding and longer-term hazards associated with seismic risk and the changing climate. Presentations of uncertain forecasts in geoscience often need to balance three communication imperatives: robustness, richness and saliency (Stephens et al., 2012). In recent years, there has been a much greater volume of geoscience research which explores this space of communication imperatives in creative ways, informed by examples from other fields (Spiegelhalter et al., 2011).

For long-term climate risk, one focus has been the narrative consistency of predictions, through for example the storyline approach to communicating climate risk (e.g. Shepherd et al., 2018; **?**). This approach can have obvious advantages for a wide variety of end users. Nonetheless, there are still many situations where communicating environmental risk forecasts that result from an ensemble of model predictions is an appropriate choice, often when there is an explicit or implicit cost-loss basis to any decision. In this study, we focus on understanding the cognitive process under which end-users might interrogate and act upon different representations of the kinds of ensemble forecasts common across many environmental sciences.

Generally, past research has shown that including uncertainty information with forecasts helps end-users make more economically rational decisions, both for non-experts (e.g., Nadav-Greenberg and Joslyn, 2009; Roulston and Kaplan, 2009; Savelli and Joslyn, 2013) and experts (e.g., St John et al., 2000; Nadav-Greenberg et al., 2008). One reason humans are able to quickly and efficiently interpret complicated probabilistic information is the use of heuristics (Tversky and Kahneman, 1974). Heuristics help simplify probabilistic information so it can be used to inform decisions. The anchoring heuristic, for example, helps a user interpret data based on a particular value (Tversky and Kahneman, 1974). For example, when negotiating a price, a person would be anchored around an asking price, if provided.

Although anchoring can help interpret information to make decisions, it can also hinder communication. For example, professional forecasters tended to forecast a higher wind speed when given model output showing maximum possible wind speeds compared with modelled median wind speeds (Nadav-Greenberg et al., 2008). Anchoring can affect interpretations of graphical data as well. On the US National Hurricane Center's hurricane track graphic, often referred to as the "cone of uncertainty", including a centre line distracted users from possible hurricane tracks away from the centre line (Broad et al., 2007). In our previous experiment using the same survey as in this study, both experts and non-experts succumbed to anchoring when making forecasts based on uncertain data for a new or unfamiliar forecast style, reporting values that were significantly closer to the value of the anchor than when no anchor was provided (Mulder et al., 2020). Additionally, anchoring lines caused non-experts to underestimate extreme values (Mulder et al., 2020).

Beyond heuristics, other design choices can affect end-users' decisions. Providing tornado warnings with probabilistic information of where a tornado might strike increased response in areas of higher risk than deterministic warnings (Ash et al., 2014). Different designs of box plots (Correll and Gleicher, 2014; Bosetti et al., 2017) or graphs (Tak et al., 2013, 2015) showing the same data can also affect user decisions and interpretations based on that data. Similarly, when forecasting maximum values from data (again, using the same survey as in this study), participants interpreted values from the box plot as between the top of the box and the top of the whisker, even though the plot's key stated that the top of the whisker was only the 90th percentile of the data. Participants interpreted maximum values greater than what was shown on the fan plot, which similarly showed

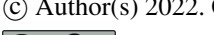



a maximum value representing the 90th percentile. Participants interpreted maxima as they were shown on the spaghetti plot (Mulder et al., 2020).

Although end user survey responses can inform on how heuristics and design choices can affect users' decisions and data interpretations, they do not always provide information on how users come to their decisions. This can make it difficult to generalise the results from experiments with end users since context and prior experience and the inherent limitations of surveying methodology limit general applicability.

In this study, we make use of another approach widely used in other fields: eye-tracking users during a decision task. Eye-tracking monitors users' eye movements during the decision making process, giving insight into how users process complex information before making a final choice (McSorley and McCloy, 2009; Orquin and Loose, 2013; Balcombe et al., 2015). Eye-tracking has been used across many disciplines such as sport (e.g. how players anticipate the landing location of a football North et al., 2009), medicine (e.g., detecting a lesion in a mammogram Kundel et al., 2007), and decision theory (e.g., investment decisions Rubaltelli et al., 2016). Eye-tracking has previously been used in environmental science to test data presentation such as colour and legends for floods (e.g., Fuchs et al., 2009) and hurricanes (Sherman-Morris et al., 2015). However, to our knowledge, eye-tracking has not been used to determine how different presentations of forecast uncertainty influence the decision making process, the focus of this study.

In particular, we seek to answer the following questions:

1. How is the use of the anchoring heuristic influenced by different types of data presentation?

2. How do people interpret uncertainty information from different types of data presentation?

3. Does data presentation affect the cognitive load required to make decisions?

By answering these questions we are able to offer guidance about how presentation choices influence the decision making process for generic, uncertain geoscientific forecasts. Although this study uses the same survey as a previous study (Mulder et al., 2020), the focus of this paper is on the cognitive process behind decisions made based on uncertain information, rather than the economic rationality of the decisions that were made.

## 2 Methods

### 2.1 Participants

A decision task survey was conducted with a range of representations of environmental forecasts showing uncertainty. Participants' eye movements were tracked while they completed the tasks. The length of time and location where their eyes focused, or fixated, on the environmental forecasts were measured. Participants were undergraduate students recruited from the University of Reading through email invitation. Participants were aged between 18–32 (mean 21.2). There were 38 females, 27 males. Participants were between 0–4 (mean 1.0) years along in their degree course. Participants were paid for participation in the study, but there was no incentive for performance on the study task.





The survey used in this study was the same as used in previous studies, which focused on the economic rationality of decisions made using different graphic types (Mulder et al., 2020). This study had a wider participant base of 121 expert (academics and professional forecasters) and 99 non-expert (decision-makers in industry and members of the general public) users. Due to the time-intensive nature of the eye-tracking study, it was not feasible to obtain a large sample representative of the wider public. However, the sample's survey responses replicated those of the non-experts in the previous studies except for maximum ice thickness interpretations for the fan plot, anchoring in the maximum ice thickness interpretations, and the difference in confidence across forecast representations (Mulder et al., 2020, see Supplementary Discussion).

## 2.2 Decision and interpretation tasks

A hypothetical scenario of ice-thickness forecasts were provided for a fictional location. Ice-thickness was chosen deliberately, as form of rarely made forecast with which participates were unlikely to be familiar, helping to control for participants' pre-conceived notions of data uncertainty in relation to a specific phenomenon. In the scenario, participants were told they were making shipments across an icy strait. Using ice-thickness forecasts, they had to decide whether to use a small ship (which could crush through ice up to 1-meter thick) or a large ship (which could crush through ice over 1-meter thick). The small ship cost £1,000 whereas the large ship cost £5,000. If the participants chose the small ship and the ice was thicker than 1 meter, there would be spoilage, resulting in an additional cost of £8,000.

Ice thickness forecasts were presented in seven different representations: deterministic line, spaghetti plot with or without a median line, fan plot with or without a median line, and box plot with or without a median line (Fig. 1). The presence of the median line was used to test the anchoring heuristic. Each of these seven forecast representations was shown to represent 30%, 50%, and 70% probability of ice thickness exceeding 1 meter. Therefore, each participant received 21 forecasts (7 representations x 3 probabilities). The order of the forecasts were randomised to reduce ordering bias.

For each forecast representation, participants completed decision and data interpretation tasks while their eye movements were recorded. Participants decided which ship to use, noted confidence in their decision (along a 10-cm visual analogue scale, 0 cm corresponding to "Not at all confident" and 10 cm corresponding to "Extremely confident"), and forecasted their best-guess, maximum possible, and minimum possible ice thickness.

## 2.3 Eye tracking apparatus

Participants were fitted with an Eye link II tracker headset (sampling rate 500Hz), which recorded eye movements of the right eye as they completed the survey on a 21-inch colour desktop PC (refresh rate of 75Hz). A chin rest was used to constrain any head movements and participants were placed in a set position. The distance between the monitor and participant was 57 cm. In this study, we monitored the location and duration of eye fixations, defined as a maintained gaze (the eye was still, i.e. no saccades were detected) on one location. For more information on methods used in eye-tracking studies, see Holmqvist et al. (2011).





## 3 Results

### 3.1 Understanding how the anchoring heuristic influences interpretation of environmental forecasts

Eye-tracking helps us to explain the anchoring seen in the survey results. Composite heat maps, which accumulate eye fixation times across all participants, indicate that although participants look at multiple possible values for ice thickness when making their ship decision (e.g., Figs. 2a,b illustrates this effect for the spaghetti plot, heat maps for the rest of the forecast representations are in the supplementary figures) and maximum ice thickness forecast (Figs. 2c,d), their eye movements were

125 anchored toward the median line, when provided. Without the median line, the eye movements tracked wider in the vertical for the spaghetti plot (Fig. 2b). Even in the maximum ice thickness forecast, where the median would not necessarily be relevant, participants still fixated on the median line (Fig. 2c).

The location of the eye fixations in choice of ship decision when given a median line was significantly closer to the location of the median line than when there was no median line (based on the absolute fixation distance from the median line). This was

130 the case for all forecast representations at all probability levels (Fig. 3). A repeated measures ANOVA showed main effects of forecast representation (F=8.82, p<0.001), presence of a median line (F=56.43, p<0.001), and probability level (F=253.30, p<0.001) on the best-guess forecast. There were no significant interacting effects. We would expect eye fixations to differ based on forecast representation (different types of representations have different foci in different locations) and probability level (the location of the information changed as the probability of exceeding 1 meter increased). Forecasts with median lines resulted

in eye fixations significantly lower (closer to the median line) than forecasts without median lines (post-hoc, Bonferroni t-test, p<0.001). Therefore, the presence of a median line affected the location of participants' eye fixations, verifying the effect of the anchoring heuristic.

Additionally, there was significantly less variance in where participants looked when there was a median line compared with no median line for the spaghetti plot at 50% and 70% (F-test; 50%: F=0.88, p=0.018; 70%: F=0.71, p<0.001) and fan plot

at all probabilities (30%: F= 0.86, p=0.007; 50%: F=0.73, p<0.001; 70%: F=0.80, p<0.001). Unexpectedly, there was more variance in where participants looked with a median line compared with no median line for the spaghetti at 30% (F=1.27, p<0.001).

### 3.2 How do people interpret information from different types of data presentation?

When looking at how a larger population of experts and non-experts interpreted maximum values from the survey used in

this study, Mulder et al. (2020) found that non-experts may be misinterpreting the box plot. The key shown with the box plot defined the top of the whisker as the 90th percentile and the top of the box as the 75th percentile. Despite the key, non-experts interpreted maximum values as between the top of the box and the top of the whisker (Mulder et al., 2020). This is a problem because box plots are commonly used in presenting environmental data. The fan plot, which similarly showed values up to the 90th percentile, encouraged maximum forecasts greater than the 90th percentile, suggesting a more rational understanding of

the plot. Participants' (both from Mulder et al. 2020 and the eye-tracking survey reported here) maxima for the spaghetti plot were not significantly different from what was shown.





One reason for participants misinterpreting maxima for the box plot is they were not referencing the key. When the first box plot was shown to each participant, the total number of seconds they fixated on the key and the number of fixations on the key were recorded (Figs. 4a,b). There was a significant main effect of plot type on the number of seconds participants fixated on the key (one-way ANOVA, F=23.03, p<0.001). Participants fixated on the box plot key (mean 7.8% of the time it took to complete the question) for significantly less time than the fan plot key (mean 17.4% of the time it took to complete the question, Bonferroni corrected t-test, p<0.001). Similarly, participants fixated on the spaghetti plot key (mean 5.1% of the time it took to complete the question) for significantly less time than the fan plot key (Bonferroni corrected t-test, p<0.001).

Additionally, there were fewer fixations on the box plot key than the fan plot key (Fig. 4b). There was a significant main effect of plot type on the number of fixations on the keys (one-way ANOVA, F=28.91, p<0.001). Participants had fewer fixations on the box plot key (mean 6.2% of fixations were on the key) and spaghetti plot key (mean 4.9% of fixations were on the key) than the fan plot key (mean 15.2% of fixations were on the key, Bonferroni corrected t-test, p<0.001 for both). With subsequent box plot forecasts, the number of fixations and number of seconds fixating on the key reduced further (not shown).

The amount of time elapsed before participants' first fixation on the key was significantly longer for the box plot than the fan plot (Fig. 4c). There was a significant main effect of plot type on the amount of time elapsed before the first fixation on the key (F=12.87, p<0.001). Participants let more time elapse before first fixating on the box (mean 7.0 seconds) and spaghetti (mean 8.1 seconds) plot keys than the fan plot key (mean 3.6 seconds, Bonferroni corrected t-test, p<0.001 for both).

Participants took longer before looking at the box plot key and looked at the key fewer times. Without referencing the key, participants were unlikely to know what each part of the box or whiskers represents for this complicated plot. Additionally, the amount of time elapsed before participants first fixated on the key was calculated. It took longer before participants first fixated on the key for the box plot compared with the fan plot. Participants may think they already understand what the symbols in the box plot represent and therefore may not reference the key for long enough to get an accurate interpretation of the plot. Alternatively, participants may simply not understand the box plot or its key.

Participants had a higher number of fixations on the fan plot key and waited less time before their first fixation on the key. Even though the fan plot may have been new to the participants, extreme values were more accurately understood in the survey results, perhaps because the forecast representation is easier to understand and they took time to study the key.

Similarly to the box plot, there was little time spent fixating on the key of the spaghetti plot, few total fixations, and longer time before the first fixation, however the survey results suggest the participants correctly interpreted the spaghetti plot. The reason for the little attention paid to the spaghetti plot could be that the type of forecast representation is intuitive or that the key was simple and therefore did not require much time to understand (Figs. 1b,c). Indeed, Bosetti et al. (2017) found that presenting climate information including individual model estimates aids in user interpretation of the data.

### 3.3 Does data presentation affect how long it takes a participant to complete a task?

One question about including uncertainty information and designing forecast representations is whether it affects the amount of time a user takes to make their decisions. This is particularly relevant when users have limited time or attention to make inferences from the data. To address this, we calculated the amount of time it took each participant to complete each task.



While the participant was getting used to the equipment and the types of questions asked, it took longer to respond to the questions. After the first two series of questions, the amount of time to complete the tasks converged to a standard average time. Therefore, the first two series of questions were removed from this analysis.

Anchoring affected the amount of time it took to complete the survey tasks (Fig. 5). There was a significant main effect of
anchoring (repeated measures ANOVA, excluding the deterministic forecast, F=20.79, p<0.001) and probability level (F=5.24, p=0.030) on the amount of time for participants to make ship decision. The interacting effects were also significant (F=4.41, p=0.040). There was a significant main effect of anchoring (F=11.85, p=0.001) on best-guess forecast with no main effect of probability and no interacting effects. There was also a significant main effect of anchoring (F=4.64, p=0.035) on maximum ice thickness forecasts with no main effect of probability and no interacting effects. There was no significant main effects of
anchoring or probability on confidence with no interacting effects.

Using Bonferroni corrected post-hoc t-tests for the above significant ANOVAs, the inclusion of a median line significantly reduced the amount of time to complete both the decision (p<0.001) and best-guess forecast tasks (p=0.002), but significantly increased the amount of time to interpret the maximum (p=0.048) ice thickness.

## 4 Conclusions

The results of the eye-tracking study verify previous research that anchoring affects non-experts' interpretations of data (Mulder et al., 2020). The change in eye movements across the graph area and key and the role of expertise is explored in more detail in the companion paper to this study. However, some general conclusions which apply to all groups can be drawn from the analysis in this part of the study. When interpreting best-guess ice thickness based on uncertainty data, the locations of the eye fixations were closer to the median line. In addition, there was less variance in where participants looked on the graphs when
determining the best-guess forecast when a median line was present. It took less time to complete the decision and best-guess tasks when given uncertainty forecasts with a median line compared with those without a median line. This evidence suggests that when a median line is present, participants' eyes are guided toward the median line and they make decisions without necessarily considering extreme values. We expect this effect to also be present for other forms of average or centre lines. This can be helpful for decisions and interpretations using values in the middle of a spread of data. In addition, it can help users
make faster decisions in a time-sensitive environment.

However, anchoring can be detrimental to how users interpret information, inhibiting their ability to assess values other than the anchor value (Tversky and Kahneman, 1974). Although the eye-tracking survey results did not find a significant anchoring effect on the value of maximum ice thickness as in Mulder et al. (2020), the presence of median lines increased the amount of time it took participants to interpret the maximum values.

From the eye-tracking participants' survey responses, participants largely correctly interpreted extreme values from the spaghetti and fan plots, but misinterpreted extreme values from the box plot. The eye-tracking data suggest that participants rarely looked at the box plot's key, fixating fewer times as well as waiting longer before referencing the key. Not looking at





the key or not looking long enough to decode the box plot may have led to misinterpretations of the plot. How eye fixations changed from early, intermediate, and later viewing periods is explored further in the companion paper to this study.

On the other hand, the survey results suggest that participants correctly interpreted data from the fan and spaghetti plots. Eye-tracking data show participants looked at the key of the fan plot, a representation participants were unlikely to have seen before, sooner, for longer, and with more fixations than the box and spaghetti plots. This suggests that although participants may not have been familiar with the fan plot, the key aided their comprehension. The spaghetti plot had a similar number of fixations and amount of time fixating at the key as the box plot, but the survey data suggest participants better understood the 225 forecast representation. Perhaps the simplicity of the representation was easier for non-experts to interpret.

First, median lines (but we expect similar effects with any other form of average or centre line) should only be used when it is helpful to guide a user's eye toward a particular value, for example average seasonal temperatures. Scientists communicating data need to be aware that when using anchoring lines, users tend to underestimate extreme values. When it is important that users focus on extreme values, for example, maximum flood levels, anchoring lines are not recommended.

Non-experts tended to misinterpret extreme values when given a box plot, even when a key was provided. This is likely to have been due to lack of familiarity with the form of presentation. However, the fan and spaghetti plots led to more accurate interpretations of extreme values and are recommended in place of box plots.

Communicating uncertain information was key to maintaining public health during the COVID-19 pandemic, but it is relevant to all sciences. Further work could focus on the types of graphic communications used during the pandemic to understand 235 how they are interpreted and understood by the public and decision-makers. The type of information communicated in this study (uncertainty around a potentially hazardous environmental event) is similar to that found elsewhere in the public sphere, for example in a weather forecast warning for extreme precipitation or future climate impacts. Because this survey was purposefully conducted with a hazard not familiar to our participants, there were no pre-conceived notions of forecast skill.These results are not constrained to any particular hazard and may be extended across other environmental data and hazards.

## Appendix A: Appendix A

Overall, the responses to the survey questions were similar to the results from Mulder et al. 2020 and 2020. As in Mulder et al. (2020), participants in the eye-tracking study exhibited clear anchoring in their survey responses. Best-guess forecasts were significantly lower (closer to the median line) when provided with a median line (paired t-test; 30%: t=-11.47, p<0.001; 50%: t=-3.01, p=0.001; 70%: t=-4.71, p<0.001). Contrary to Mulder et al. (2020), where anchoring produced significantly lower 245 maximum thickness interpretations, participants in the eye-tracking experiment only had significant main effect of probability on maximum ice thickness interpretations (ANOVA, F=11.42, p<0.001) with no main effect on the presence of a median line and no interacting effects. The maximum thickness increased with increasing probability (post-hoc Bonferroni corrected t-test, p<0.001 for all).

There were no significant main effects for forecast representation on best-guess forecast, but there was a significant main 250 effect of probability (ANOVA, F=19.01, p<0.001) with no interacting effects. Participants' best-guess forecasts increased with





increasing probability (post-hoc Bonferroni corrected t-test, p<0.001 for all). This was the same finding as with non-experts from Mulder et al. (2020).

There was a significant main effect of forecast representation (ANOVA, F=5.50, p=0.004) and probability (F=7.66, p=0.006) on confidence with no interacting effects. Forecast representations' effect on confidence became insignificant in post-hoc tests
(Bonferroni corrected t-test). That forecast representations had no effect on confidence between the spaghetti, box, and fan plots corresponds to the results from Mulder et al. (2020). Unlike Mulder et al. (2020), the eye-tracking participants chose the large ship more significantly frequently as the probability of exceeding 1 meter thickness increased (p<0.001 for all).

In Mulder et al. (2020), there was a significant difference in confidence by probability and no significant difference by forecast representation for non-experts. Conversely, with the eye-tracking participants, there was no significant main effect of
probability and a significant main effect on confidence (F=7.90, p<0.001) with no interacting effects. Confidence in decisions was significantly greater for the fan plot than the box and spaghetti plots (post-hoc Bonferroni corrected t-test, p<0.001 for both).

For maximum ice thickness, eye-tracking participants consistently predicted maxima less than the top of the whisker (t-test; 30%: t=-5.80, p<0.001; 50%: t=-7.68, p<0.001; 70%: t=-5.40, p<0.001) and greater than the top of the box (30%: t=5.95,
p<0.001; 50%: t=6.89, p<0.001; 70%: t=6.96, p<0.001), similar to the results in Mulder et al. (2020). Contrary to the findings in Mulder et al. (2020), the eye-tracking participants forecast maximum ice thickness significantly greater than what was shown for the fan plot only at 30% (t=2.84, p=0.003). For the fan plot at 50% and 70%, there was no significant difference between the participants' interpreted maximum ice thickness and what was shown as the maximum value on the forecast. Similar to the findings of Mulder et al. (2020), there was no significant difference in maximum ice-thickness interpretation and the maximum
shown on the spaghetti plots at 30% or 50%, but was less than the value shown on the spaghetti plot at 70% (t=-2.71, p=0.004).
      :

*Author contributions.* Kelsey Mulder: Formal analysis, Visualization, Writing – original draft preparation Louis Williams: Conceptualization, Investigation, Writing – review  editing Matthew Lickiss: Writing – review  editing Alison Black: Funding acquisition, Writing – review  editing Andrew Charlton-Perez: Funding acquisition, Writing – review  editing Rachel McCloy: Funding acquisition, Writing –
review  editing Eugene McSorley: Conceptualization, Resources, Writing – review  editing

*Competing interests.* No competing interests are present

## Ethical Statement

The University of Reading Ethics Board approved the study, and the study was conducted in accordance with the standards described in the 1964 Declaration of Helsinki. Participants provided written informed consent. The authors declare that there
is no conflict of interest.





*Acknowledgements.* We thank our eye-tracking study participants. This research is funded by the Natural Environment Research Council (NERC) under the Probability, Uncertainty and Risk in the Environment (PURE) Programme (NE/J017221/1). Data created during the research reported in this article are openly available from the University of Reading Research Data Archive at http://dx.doi.org/10.17864/1947.110.



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





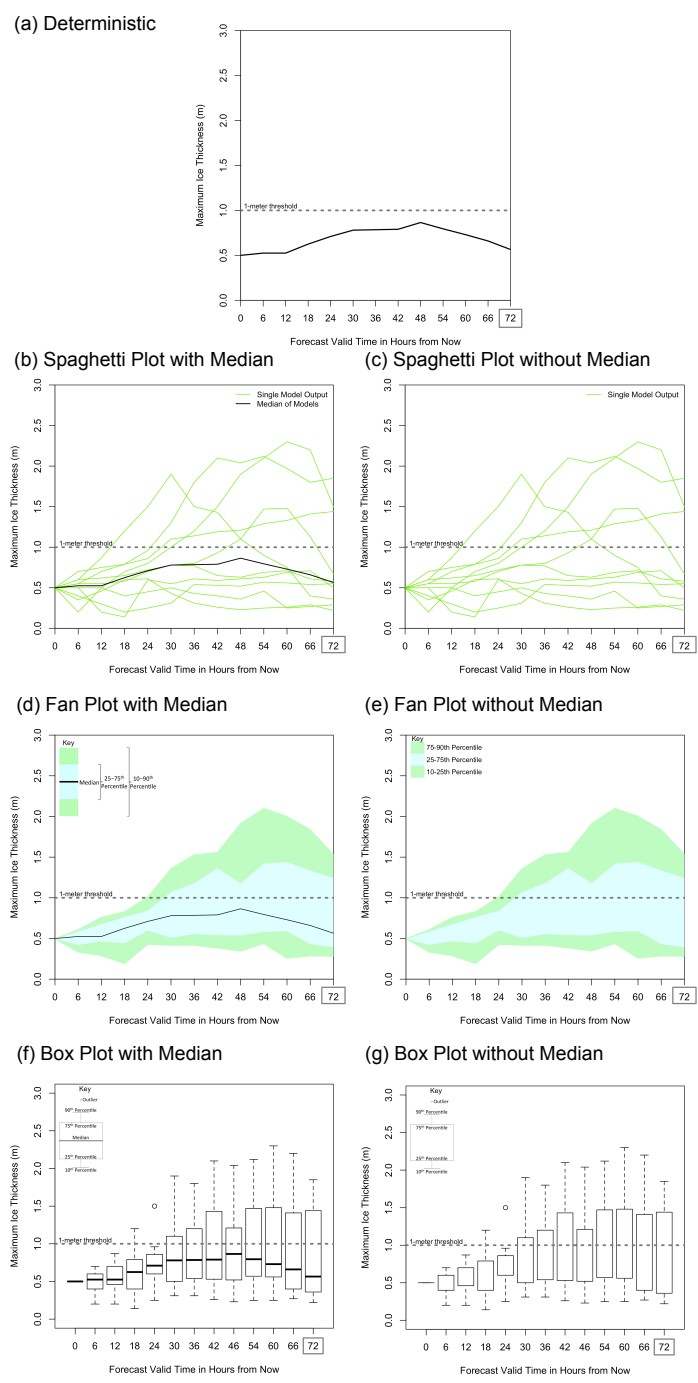

**Figure 1.** The four forecast representations used in this analysis: (a) deterministic (using only the median line), (b) and (c) spaghetti plot, (d) and (e) fan plot, and (f) and (g) box plot. Uncertainty forecasts were shown both with median lines (b,d,f) and without median lines (c,e,g). All forecasts represent the same information: three of 10 model runs show ice greater than 1-meter thick. The same plots were produced for 50% and 70% chance of ice greater than 1-meter thick (not shown). The dotted line in each graphic shows 1-meter ice thickness, the threshold the participants predicted.





**Figure 2.** Composite heat maps accumulating the duration of eye fixations (in milliseconds) of all participants for the ship decision (a,b) and maximum ice thickness (c,d) tasks. Heat maps are shown only for the spaghetti plot with (a,c) and without (b,d) median lines. Heat maps for the other forecast representations can be found in the Supplementary Figures. Between each question, there a cross was present to help participants focus back to to the centre of the screen prior to moving on. Artefacts of this centring can be seen on the heat maps.





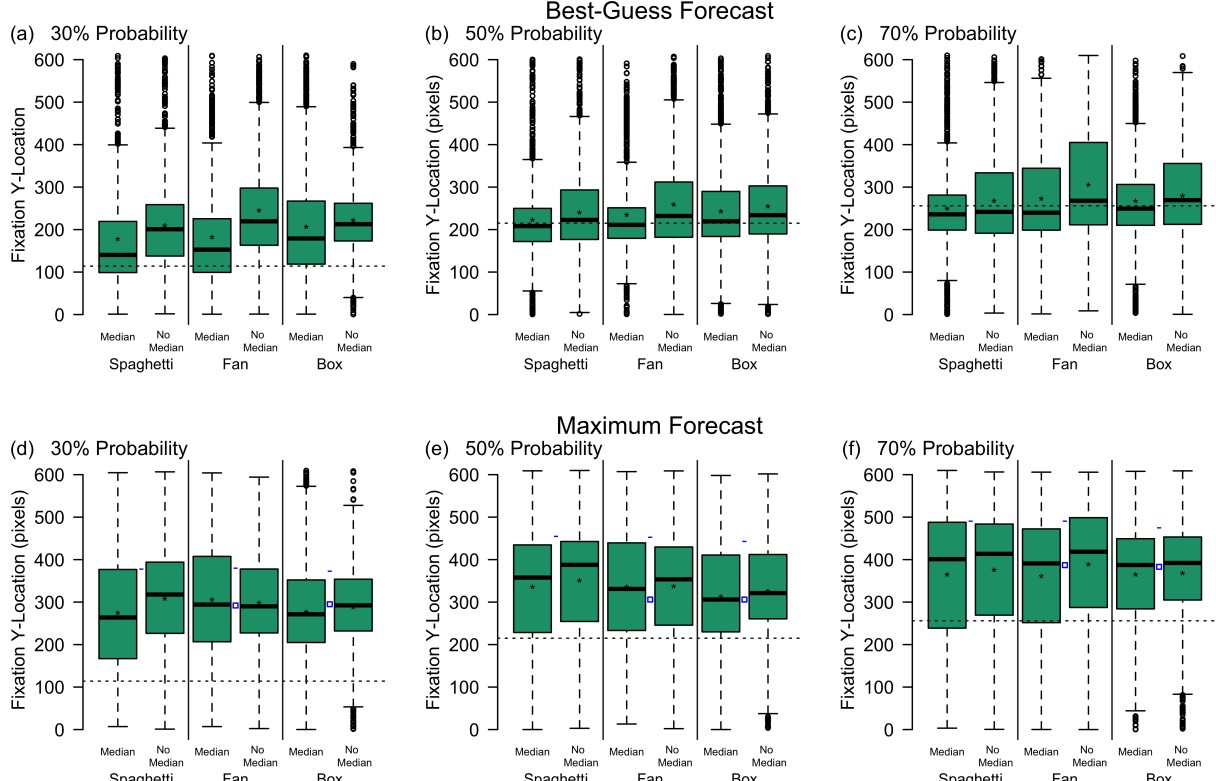

**Figure 3.** Y-Axis fixation location when asked to provide their best-guess ice thickness (a,b,c) or maximum ice-thickness forecast (d,e,f), given a 30% (a,d), 50% (b,e), or 70% (c,f) probability forecast. Results are separated based on forecast representation (spaghetti, fan, and box) and whether or not a median was present. Stars represent the mean. The dashed line shows the median value, shown in the line representation and representations with a median line. In (d,e,f), the blue dashes represent the maximum ice thickness value shown for the spaghetti, fan, and box plots. The blue squares show the top of the 75th percentile value for the fan and box plots.





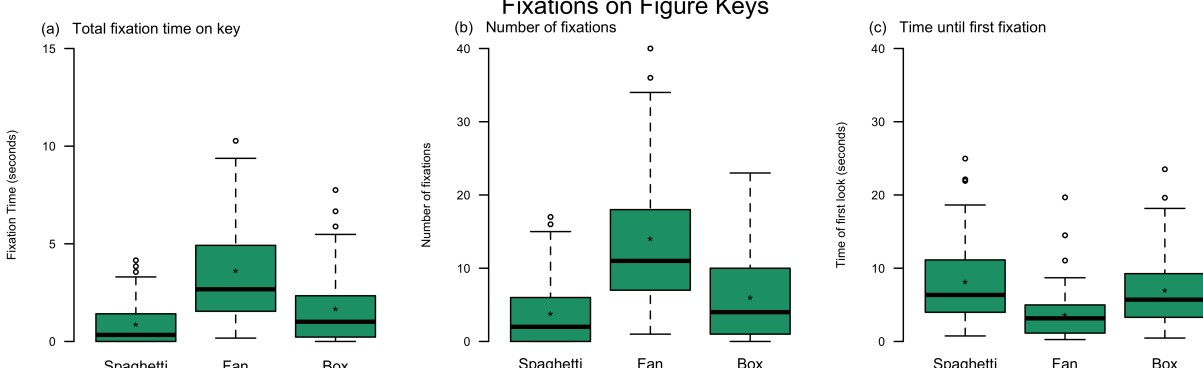

**Figure 4.** Eye tracking analysis of a participants' first look at each graphic type to determine (a) the total amount of time spent looking at the key, (b) number of fixations within the key area, and (c) amount of time from the beginning of the question it took for the participant to look at the key.







**Figure 5.** Amount of time it took each participant to complete each question: (a) ship decision, (b) confidence in decision, (c) best-guess ice thickness, and (d) maximum ice thickness. The first two graphics each participant used were removed from this analysis to remove trials where participants were getting used to the task. After two graphics, the amount of time it took participants to complete each task converged.







**Figure 6.** Supplementary Figure S1: Heat maps overlaying the eye fixations of all participants for the ship decision for the deterministic line (a), box plot (b,c), and fan plot (d,e). Heat maps are shown with (a,b,d) and without (c,e) median lines. Between each question, there a cross was present to help participants focus back to to the centre of the screen prior to moving on. Artefacts of this centring can be seen on the heat maps.





**Figure 7.** Same as Fig. 6, but for maximum ice thickness.