# Peer review of "Understanding representations of uncertainty, an eye-tracking study part I: The effect of anchoring"

_EGUsphere, 2022_

## Author Comment (AC1)

Response to RC1:

This is an important issue - not just for users, but for professionals formulating warning messages. Anchoring can be positive, e.g. if the anchor line is set at an agreed response threshold e.g. reasonable worst case, or for a specific cost/loss ratio, but it can also be negative especially in a low probability high impact situation, if highlighting of the median leads to underestimation of risk. Thanks for your comment.

Line 84: It should be noted that the sample is not representative of typical users. I don't believe this undermines the results in any way, but a comment on the differences between the sample and typical real life users might be worth including. We have noted that these participants are not necessarily representative of all populations

Line 105-6: The meaning of the three probability levels is not clear. I believe it may mean that there were three forecast scenarios presented with data adjusted to give 30%, 50% and 70% probabilities of exceeding 1 metre, and that each was presented in 7 different ways. However, I am still not sure if that is a correct interpretation. There is no figure that shows what the three scenarios look like (maybe the deterministic presentations would clearly display them?). I think it would also be helpful to include a brief description of how the 3 scenarios were created. Your interpretation is correct. We have reworded to make this clearer. For conciseness, we have not included depictions of the 50% and 70% probabilities but have included a description for how these were created (we used the spaghetti plot as a base and had 3, 5, and 7 forecast lines above the 1-meter mark, respectively. Then we converted the spaghetti plot into the other plot types).

Line 131: The authors should not assume that all readers will be familiar with the fit statistics for the ANOVA or Bonferroni tests. I suggest adding a short section 2.3 to introduce the statistical tests used and the meaning of the fit statistics (supplemented by a suitable reference). Thanks for this comment. We will add an explanation of the statistics used in the updated paper.

Line 179: The explanation of the reason that good interpretations were made of the spaghetti plots despite even less attention to the key than for the boxplots may be correct, but it runs counter to the general view that spaghetti plots should be avoided because they are difficult to interpret (due to crossing lines etc). I think this view should be acknowledged and responded to. We have cited a paper that found that climate information including individual model estimates helps user interpretation of data, which is counter to the general view you cite, however, we are happy to include reference to this in the manuscript

Line 189: This paragraph does not tell us what the impact on answer time actually is, merely what the statistics are of the differences. We should not have to look at the figure to work this out. We have updated the manuscript to make sure that the direction of the effects reported is stated explicitly in the text.

Line 198: a comment on the increase in time to estimate the maximum in a spaghetti plot with median is needed. This analysis was conducted combining all forecast representations together, not conducted separately for each forecast representation, like in section 3.2. Therefore, a comment specifically on the spaghetti plot would not be appropriate.

Line 211: This line starts with "However", which seems out of place given that the previous paragraph was also talking about the dangers of anchoring. Thanks for pointing this out. We've amended the manuscript to remove the "however".

---

## Author Comment (AC2)

Response to RC2:

General Comments: I have major concerns with this manuscript as written. First and foremost, I find that there's a mismatch between how the results are reported and what is described in the methods. The author's stated focus in this article is on the results of an eye-tracking study of different uncertainty visualizations, but the authors also refer at times to a survey that was conducted along with the eye-tracking results. As it stands, however, the authors barely describe this survey in the methods and don't provide any of the survey results in the paper proper (with a brief discussion tucked away in the appendix). These results are necessary to contextualize the eye-tracking results, and so the result is a paper that feels half done.

Thank you for these comments. This paper is a stand-alone paper that reports the outcomes from a completely separate analysis of an eye-tracking study on different forecast representations. This used the same survey tool employed previously (Mulder et al, 2020) but examined eye-tracking responses to different forecast representations used. We do reference the values of the survey results in the appendix on occasion, where appropriate, but only as support for the main focus of the paper – the eye-tracking outcomes. We have modified the manuscript to make this clearer.

There are other major issues to contend with as well – the authors frame this article in terms of COVID-19, when it is not clear how these uncertainty visualizations are related to COVID-19; there are some spots in the intro where the authors do not provide enough supporting literature; and there are organizational issues throughout the paper that make it difficult to follow. The eye-tracking results, depending on how they are framed, could be interesting, especially when paired with appropriate survey/decision task analysis, but as currently written, I believe the authors need to go back to the drawing board.

Thanks for these comments. The reference to COVID-19 was intended to contextualise how uncertainty visualisations are increasingly being used, even outside weather and climate uses as the study was originally planned to focus on. We want to impart that researching how uncertainty visualisations are understood is important. We have revised the manuscript to make this clearer. As for your other comments, they are addressed individually below.

Major Comments:

Lines 3-4 and 17-19 – I'm confused by the reference to the COVID-19 pandemic. Are the visuals you study in this analysis commonly used to visualize COVID? Just seems like this framing doesn't really fit, especially when the rest of the intro is focused on applications in geosciences We have reworded the abstract and introduction to clarify that communication of uncertainty is not just a geosciences problem, that COVID-19 put uncertainty communication in front of the world and that this problem is universal, not specific to geosciences.

25-27 – feels like there should be a few more citations here, if there has truly been a "much greater volume of geoscience research" on these topics We have added additional citations to support this statement in the manuscript

30-31 – unclear what this means – implying that narrative consistency approach is incompatible with ensemble predictions? We understand how a reader could be confused by this statement. We have clarified within the manuscript to explain that sometimes the narrative changes with evolving science or model forecasts and that change in narrative can lead to distrust in the forecast.

50-60 - this paragraph feels a bit garbled…not sure what the key takeaway is here. One thing I might suggest is devoting an entire paragraph to the previous research from your research group – splitting it between two paragraphs breaks up the flow. And this might make it easier to establish what your previous research has already established, and what this study is trying to learn to build on those results. We have taken your advice and split this paragraph into two to clarify the previous work outside our group and the research within our group and how this study builds upon it.

56 – it might be helpful to explain/provide examples of the types of visuals you look at in more detail somewhere in the introduction. Fan plots, for instance, are not something I was familiar with before reading this. We have explained the appearance of a fan plot and referred the reader to our example in Figure 1 d, e.

70-71 – might cite Sutton & Fischer (2021) or Sutton et al (2020) here or elsewhere in the intro – not related specifically to your topic of study, but provides some previous examples of eye-tracking in geoscience communication Thanks for providing these references. We have read these articles and include them in our citations in the revision.

72-75 – some thoughts on these RQs. First, I think it would be useful to define the various types of data presentations you're interested in looking at (e.g. box plot, fan plot, spaghetti plot with/without medians) before leading in here. At this stage, it's unclear what is meant by "data presentation". We agree that this is certainly one way to present the research questions, however we feel the definition of the data presentations is better placed in the methods section. To clarify what we mean by "data presentation", we have replaced that phrase with "presentations of forecast uncertainty" to reflect the sentence before the research questions.

Second, the RQs are too vague and mismatched with your results. The first section of results focuses on how the presence of a median line affects where participants eyes go when making decisions, and so I might reframe this question to be more in line with how central tendency lines affect visual processing of risk visualizations. Likewise, the second set of results focuses on how participants engage with the information in the key of the visuals – I'm not sure how this is related to "uncertainty information", except in a very broad sense. The third RQ is better, but I might be even more specific in asking whether participants take more time to make decisions based on visual attributes – it's just hard to tell from these results whether the time taken to make a decision is due to cognitive load or some other factor. Also, I'm not sure this final RQ is really supported by the literature provided in the introduction. Why is this concept important?
We have reworded the research questions in the manuscript to be more specific, aligned with your suggestions. They now read:

1. How is the use of the anchoring heuristic (in this study, tested using a central tendency line) influenced by different presentations of forecast uncertainty?
2. How do people interpret uncertainty bounds (minimum and maximum possible outcomes) from different presentations of forecast uncertainty?

3. Do different presentations of forecast uncertainty affect the amount of time required to make decisions?

84-87 – might lead off this paragraph with the description of your participants, then talk about measures Because we have not yet explained that this is a decision task survey, we decided to give a quick overview of what the study is before talking about the participants.

88-94 – this paragraph feels a bit misplaced – perhaps it should be included in the next section, when you talk about the decision tasks that were part of the survey?
See above

This paragraph is also unclear because you say that you used the same survey instrument as in a previous study, but also mention that the previous study had a different focus. Does this mean that you also asked about economic rationality of decisions in this study, and didn't report the results from those questions? Did you also use eye-tracking in the previous study? Was the experimental method the same between the two studies, just with a different population? It sounds from the way you phrase this that the previous study did not use eye-tracking as part of the experimental method. If this is the case, then I'm not sure the replication information is useful, considering the varying conditions in which the participants were answering the question. We have clarified this to say that while the same survey materials were employed, eye tracking was additionally used in this study. This enabled us to examine the process by which decision were made whereas the previous survey study (Mulder et al., 2020) was more focused on the outcomes of the decisions themselves.

95 – do you make it clear in this section that participants are asked to make a decision based on a 72-hour forecast? Why this forecast lead-time? We edited the manuscript to state that this is a 72-hour forecast, which was an arbitrary lead time, but that forecasts showed that ice thickness at this lead time was uncertain.

111 – I'm curious why you mention the minimum possible decision task, as it doesn't seem that you report on it here. More broadly, I think you can make the decision tasks more clear, and maybe explain why you chose those decision tasks (e.g., what are they operationalizing? Why are these tasks important? What do they elucidate?). Details about how the decision task survey was administered would be useful as well. We have edited the manuscript to clarify all the above questions. The decision itself is which boat to send, the forecast is their deterministic guess based on the forecast type. The min and max ice thickness are to determine the perceived range of uncertainty in the forecast. These were analysed in more detail in our other published paper, which is now referenced in this part of the paper as well for readers who want more information on that study.

112 – I think it would be useful to define your dependent variables in this section – e.g. you talk in the results about the duration of fixation, the time taken to look at various elements, etc., and it might be helpful to lay these out and explain what they represent and why they're important here We have included this information in section 2.3 and expanded it to more clearly explain how eye fixations and maintained gaze were used to answer the research questions.

121 – You lead off the results by noting that eye-tracking helps to explain anchoring seen in the survey results, but you haven't presented any survey results. What results are you talking about

then? Are these results from a previous survey? The results presented in the appendix? A forthcoming publication? As you note here, these survey results are essential to understand and interpret the eye-tracking results. Agreed. We have added to the manuscript a citation to the previous study and explained that the deterministic forecast was anchored to the central tendency line, when provided.

123-124 – there's too much information loaded into the parentheses – maybe note somewhere earlier in this section that you only visualize results for the spaghetti plot and that other visuals are in supplementary information? But I guess this raises a more important point – why did you only choose to highlight the spaghetti plot visuals? Agreed. We have attempted to reduce the parentheses. We have also noted the visualization of the results earlier in the section as suggested. The decision to include only one visualization in the main manuscript was made due to length and clarity. The others, as we note are available as supplementary material. There was no special reason to include the spaghetti plots. They just happened to be style that was chosen.

128 – another related concern – in this section, you talk exclusively about the ship decision and maximum ice thickness forecast decision tasks. Where are the results for the other decision tasks asked of participants? We mentioned in the methods section that economic rationality (the decision part of the decision task) is discussed in another paper (with citation provided – Mulder et al, 2020), and for that reason we did not include those results in this paper. The minimum ice thickness was skewed to 0 because of the nature of the forecast (it cannot be less than zero) so these results were omitted from analysis. This is now discussed in the methods section.

128-130 – I don't see ship decision in Figure 3? Unless ship decision just means best-guess forecast, but it seems like those were different tasks This should refer to Figure 2, which has been updated in the manuscript.

138-142 – I would question how practical some of these results are. For instance, the median vs no median variance for spaghetti at 30% doesn't appear to be significantly different Please see the next paragraph in the manuscript for a breakdown by probability.

143 – this section header seems inappropriate – this section is mostly focused on legend fixations for the various data presentations, so I would be more specific. "Interpretations" is a bit too broad for me. Changed header to "How do people interpret uncertainty bounds from different presentations of forecast uncertainty" to match our changed research question, discussed above.

143 – general question for this section – are these results for one decision task or for all decisions aggregated? Would be helpful to know Results in this section are aggregated across probabilities, but calculated separately for each forecast presentation type. The manuscript now explicitly states this.

144-151 – this discussion feels misplaced here, considering these are results from a previous study which should have been covered in the introduction. It gives the impression that these are results from this study, but this doesn't seem to be the case. Removed "from the survey used in this study" to make it more clear that we are referring to a past finding from Mulder et al., (2020) where non-experts may be misinterpreting the box plot.

150-151 – clarify this sentence – are you saying that the maxima provided by participants was close to the highest member of the spaghetti plot? Yes. We clarified using your language – the highest member of the spaghetti plot

152 – passive language. Also, based on what? This is a possible assertion, but it is not proven by any of the results that follow Changed to clarify that one hypothesis for participants not understanding the box plot is that they are not looking at the key. And we explicitly state this at the end of the paragraph to say our eye tracking results support this hypothesis

152-153 – from the results that follow, it seems that you recorded the number of seconds fixating on the key and the number of fixations on the key for all image types, not just for the first box plot. So this sentence needs to be clarified. Clarified by adding "the key of each plot"

162-163 – clarify here – do you mean that as the participants were exposed to additional box plots as part of the experimental design, they fixated less on the key? This raises additional questions about how the results in this section are reported – do the mean fixations and time spent fixating on the key include every time the participants viewed the specified visualizations, or just the first time? Also, do you see a similar pattern of fewer fixations/time spent fixating on the key for the other visualization types, or just box plot? We have clarified this paragraph to make it clear we're only talking about the spaghetti plot here. We believe it's clear in this section which metric we're referring to. For example "The total number of seconds they fixated on the key" and "number of fixations on the key". But agree it is less clear in the final paragraph, which we have amended for clarity.

168-181 – these last three paragraphs again reference the "survey" results which are not reported elsewhere and which are needed to make sense of the eye-tracking results We have clarified that we are referring to the survey given with the eye tracking study, which is the same survey as cited elsewhere and explicitly stated which tasks we're referring to, for clarity.

170-171 – sentence beginning "It took longer before…" – isn't this just rehashing what is stated in the first two sentences of this paragraph? Agreed. Removed this sentence.

183-185 – great lead into this section, but is this question really asked anywhere in the intro (outside of the research questions)? This question is both of practical importance and supported by a wide range of literatures showing the impact of different modes of information presentation decision times. We have added reference to literature in the manuscript.

187-188 – what is meant by "first two series of questions"? Were any of the probabilities or visualization conditions overrepresented among these first two series? Also, do you take this adjustment period into account for the other analyses reported in earlier sections? The same series of questions (ship decision, best-guess forecast, maximum ice thickness, confidence in response) were used for each forecast representation. The first two forecast representations took longer before converging to an average completion time. We removed the first two forecast representations because they skewed the overall results. We have clarified this explanation in the manuscript to reflect this.

189 – unclear on first read that "anchoring" here refers to including the median line – might be better to be specific that including the median line affected the amount of time needed to make a decision Agreed. We have clarified this in the manuscript.

198 – the previous sections had some level of discussion intertwined with the results – here, that's missing and it's really needed. What do these results mean practically? How should they be interpreted? We've added this to the manuscript.

191-192 – interacting effects of "anchoring" and probability level? Would be helpful to explain what this interaction means practically We have clarified that the impact of anchoring increased as the probability level increased.

193-194 – the results reported here (significant main effect of anchoring on maximum forecast) is another example where the results may be significant but do not appear practically significant on the plots We have clarified that the significant main effect is when aggregated across plot types, which is different to what's shown in Figure 3.

215-225 – again, you mention survey responses despite not providing the survey results in this paper. And as is clear here, these results are essential to making any sort of conclusions from this data. We have clarified to state that the other survey found a significant anchoring effect on the value of maximum ice thickness, which this study did not also find, however we did find it increased the time it took to interpret maximum values. We removed references to the "survey" and instead point to this study to reduce confusion.

226 – "first"? Feels like some transition to "recommendations" is missing here We have removed "first" and also added a transition to recommendations.

227 – curious about average seasonal temperatures as an example here, and can't help but feel that a better example could have elucidated the concept more clearly. For instance, you could note that these median lines might be helpful in hurricane track forecasts where confidence in the track is high. Thanks for that recommendation. We have replaced our example with yours.

234-235 – again, curious about the COVID framing here Clarified why we reference COVID again here – that uncertainty communication is not limited to geosciences.

237-238 – these last two sentences feel like an odd way to end the paper. I think these points might be better included elsewhere, and then close the paper with the key takeaway message. Agreed. We moved this paragraph to the beginning of the conclusions section.

240 – maybe this is where you talk about the survey responses? But nonetheless, my point stands because these results need to be in the paper proper, and not tucked away in an appendix. I would pull out the survey responses from this study and place them in the actual paper, and then maybe you could leave the comparison between studies in the appendix We believe the survey results should live in the appendix because those results are really secondary to the purpose of this study where the eye-tracking results are the focus. We only include this here should interested readers wish to compare to the results from the previous large-scale survey reported in Mulder et al., (2020).

Figure 2 – I'm wondering if the information in the caption about centering should be included somewhere in the methods. Agreed. We added to the methods section.

Figure 6 – really feel that these figures should be in the main paper – the patterns of fixation helpfully illustrate the patterns explored in section 3.2, in many cases more so than the box plots provided in Figure 3 We wanted to limit the number of figures in the main paper so as to not overwhelm the reader. We felt that these plots were similar in findings to Figure 2 that they were not necessary to include in the main paper, but useful to include as supplementary figures.

Figure 7 – is there any explanation for the difference in fixation on the key for the median fan plot but not for the no median fan plot? Is this an ordering effect? The effect of median line on looking at the key was not part of this study so therefore was not discussed. Perhaps it will be included in future work.

Minor Comments/Grammar/Typos:

20-22 – grammar issue - there's a noun missing before "runs" added "scientists"

29 – misplaced question mark in citation? This was an error in LaTeX compiling, it's now been fixed.

59 – awkward phrasing…remove one of the "can"s Amended in manuscript

62 – "context and prior experience and the inherent limitations" – rephase to add commas Amended in manuscript

66-68 – add some sort of punctuation between the annotation and the references in parentheses (e.g., detecting a lesion in a mammogram; Kundel et al., 2007) Amended in manuscript

96-98 – first two sentences of this paragraph are worded awkwardly; revise Amended in manuscript

107 – "were" should be "was" Amended in manuscript

150 – the information in the parentheses should be added to the end of the sentence Amended in manuscript

177-180 – "reason for the little attention paid" – first two sentences of this paragraph use passive language; revise Amended in manuscript

178 – semi-colon needed before "however" Amended in manuscript

191 – "to make ship decision" – revise Amended in manuscript

228-229 – awkward phrasing with all the comma splices Unable to find the comma splices you're referring to.

241 – Mulder et al 2020 and 2020 – is this referring to two papers? This was a typo. Amended in manuscript

Figure 1 – the text in the graphics (especially the axis ticks, labels, and annotations) is quite small and hard to read…would encourage making it larger Agreed. We will work with the editorial team to make sure this figure is legible.

Figure 2 – "between each question, there a cross was present" – grammar, fix Amended in manuscript

Figure 2 – "centring" should be "centering" Amended in manuscript

Figure 3 – there's a LOT going on here – I think some of the smaller features (e.g. the stars as means, the blue squares and lines) are really difficult to see Agreed. We will work with the editorial team to make sure this figure is legible.

Figure 5 – the annotations in the caption (e.g., b) confidence in decision) don't line up with the annotations in the plot (where d is the confidence plot) Amended in manuscript